# A Multi-Analytical Approach on Silver-Copper Coins of the Roman Empire to Elucidate the Economy of the 3rd Century A.D.

**DOI:** 10.3390/molecules27206903

**Published:** 2022-10-14

**Authors:** Giovanna Marussi, Matteo Crosera, Enrico Prenesti, Davide Cristofori, Bruno Callegher, Gianpiero Adami

**Affiliations:** 1Dipartimento di Scienze Chimiche e Farmaceutiche, Università degli Studi di Trieste, 34127 Trieste, Italy; 2Dipartimento di Chimica, Università degli Studi di Torino, 10125 Torino, Italy; 3Centro di Microscopia Elettronica “Giovanni Stevanato”, Campus Scientifico, Università Ca’ Foscari Venezia, 30172 Venezia Mestre, Italy; 4Dipartimento di Scienze Molecolari e Nanosistemi, Università Ca’ Foscari Venezia, 30172 Venezia Mestre, Italy; 5Dipartimento di Studi Umanistici, Università degli Studi di Trieste, 34124 Trieste, Italy

**Keywords:** archaeometry, roman coins, µ-EDXRF, ICP-MS, silver alloys, trace elements

## Abstract

In this study, 160 silver-copper alloy denarii and antoniniani from the 3rd century A.D. were studied to obtain their overall chemical composition. The approach used for their characterisation is based on a combination of physical, chemical, and chemometric techniques. The aim is to identify and quantify major and trace elements in Roman silver-copper coins in order to assess changes in composition and to confirm the devaluation of the currency. After a first cataloguing step, μ-EDXRF and SEM-EDX techniques were performed to identify the elements on the coins’ surface. A micro-destructive sampling method was employed on a representative sample of the coins to quantify the elements present in the bulk. The powder obtained from drilling 12 coins (keeping the two categories of coins separate) was dissolved in an acidic medium; heated and sonicated to facilitate dissolution; and then analysed by ICP-AES and ICP-MS. The two currencies had different average alloy percentages; in particular, the % difference of Ag was about 8%. The other elements were found in concentrations <1 wt%. Of these, the element highest in concentration were Pb and Sn, which is in agreement with the literature. The multivariate analysis performed on the data acquired revealed two groups of coins, corresponding to the two currencies.

## 1. Introduction

The investigation of the chemical composition provides the fineness of ancient coins that allows to gain information on the evolution of coinage over the centuries (if samples from various periods are available and can be analysed) [1]. Coinage, in turn, can reveal politically, socially, and economically relevant facts. Moreover, it can help to understand the economic choices of the imperial authorities, especially the military ones, in times of crisis resulting from conflicts between legions, but also from the reduced supply of metal in contrast to a greater demand for metal to be minted, especially silver during the period in which the coins were issued.

The aim of this study is to identify and quantify major and trace elements in Roman silver-copper coins of the 3rd century A.D., in order to assess changes in composition between denarii and antoniniani (introduced by emperor Caracalla in 214 A.D.) and to confirm the devaluation of the currency during the “Military Anarchy” or “Imperial Crisis” (A.D. 235–284). The later 3rd and early 4th centuries A.D. were signed by a deep political and military crisis, which strongly affected the entire administrative and military organisation of the Roman Empire. From the monetary reform of Augustus (23 B.C.) and Nero (54 A.D.) to at least the reform of Caracalla (ca. 214 A.D.), the silver coin, i.e., the denarius, had played a central role in the monetary system of the Roman Empire. Its stability, however, had been weakening overtime. The silver coin adulteration, in fact, had become the remedy to which the emperors resorted in cases of increased expenses (especially military). In other words, they increased public expenditure by increasing the production of the monetary stock; they decreased the silver fineness, but the nominal value of the coin remained unchanged. In fact, from the denarius by Nero with a silver fineness of about 93.5% and a weight of 3.4 g, it came to the denarius minted between Septimius Severus and Caracalla with a silver fineness even lower than 50% and a weight lower than 3.4 g [2,3,4].

This devaluation led to increasing in the alloy the copper content, or lead and tin in some cases (base metals), and to developing new metallurgical techniques of surface silvering (a trick to mask reality of the bulk) [3]. As a consequence, the emperor Caracalla introduced a new currency, the antoninianus. This coin has been valued at 2 denarii, as indicated by the radiated crown on the recto, although it was equal to 1½ denarii in weight (5.11 g) and contained about the same percentage of silver as only one denarius. It was introduced in 214 A.D., but its continuous and massive production began only in 238 A.D., when it replaced the denarius definitely. Starting from the year 238 A.D., the antoninianus shows a deterioration in comparison to the similar currencies minted during the empire of Caracalla; indeed, the silver content dropped from about 50% to 42% and remained almost constant until about 250 A.D. The devaluation reached its peak with the last issues of the emperors Gallienus (260–268 A.D.) and Claudius II Gothicus (268–270) when the antoninianus was coined with an alloy of argentiferous bronze containing less than 5% silver [2,5].

The analytical determination of the composition of ancient coins can give a reliable documentation for historical, numismatic, and archaeological studies, providing information on ancient minting methods and contaminations introduced during manufacture but, above all, the intentional adulteration of the metal alloy of monetary issues. In addition, it allows the identification of possible corrosive processes affecting the surface, especially in the case of silver-copper alloy coins, and such information is crucial to identify conservation and restoration treatments of ancient coins [6,7].

As this type of specimen is usually very rare, maybe unique, and of great historical and economic value, non-destructive analysis is preferred. The most common non-destructive techniques for elemental determination are μ-EDXRF (micro-energy dispersive X-ray fluorescence, henceforth μ-XRF) spectroscopy, energy dispersive X-ray microanalysis in a scanning electron microscope (SEM-EDX), neutron activation analysis (NAA), and the X-ray emission induced by particles (PIXE) [6,8]. The most widely used non-destructive technique for the analysis of ancient metal alloys is undoubtedly μ-XRF, as it is rapid, inexpensive, reproducible, and semi-quantitative, at least for non-corroded and well-preserved samples [9].

Depending on the production criteria and environmental conditions to which silver-copper alloy coins are subjected, alterations to the alloy and the formation of surface concretions can occur [10]. As highlighted by Hrnjić et al. [11], copper diffusion at the surface can take place, resulting in the formation of insoluble oxidized compounds such as cuprite (Cu_2_O) and tenorite (CuO), or original intentional or post-depositional enrichment of silver at the surface may occur. In these cases, the coins’ surface is not representative of the bulk, therefore surface analysis of Ag and Cu concentrations would lead to inaccurate results, thus necessitating micro-destructive analysis to determine the alloy composition in the bulk.

In this study, resulting from a collaboration of various public research bodies, a total of 160 silver-copper based coins from a hoard found allegedly in the Balkan area, now kept in the Numismatics Laboratory of Department of Humanities (University of Trieste, Italy), was studied. In particular, they were 138 denarii and 22 antoniniani from the 3rd century A.D. (Roman emperors from Septimius Severus to Marcus Aurelius Probus) (Table 1). First, weight, diameter, and thickness of each coin were determined. Then, non-destructive techniques μ-XRF and SEM-EDX were used to identify the elements on the surface of the coins (Ag, Cu, Pb, Sn, Ni, Zn, Bi, Fe, and Ca). A micro-destructive sampling method—based on acquiring a bulk sample by mechanically drilling the edge of each coin [12]—was then used. Subsequently, inductively coupled plasma atomic emission spectroscopy (ICP-AES) and inductively coupled plasma mass spectrometry (ICP-MS) were employed in the determination of Ag and Cu (the two alloying elements), as well as of Pb, Sn, Ni, Zn, Bi, Fe, and Ca (minor elements, overall lower than 1% of the mass), in the powder resulting from the drilling of 12 coins (see Table 2), previously dissolved in a concentrated nitric acid aqueous solution, heated, and sonicated to facilitate the dissolution. The samples for the micro-destructive analyses were chosen because they had greater average thickness, also paying attention to currency, so that the different compositions could be compared.

## 2. Results and Discussion

### 2.1. Qualitative Analyses

#### 2.1.1. µ-EDXRF Results

As the depth analysed is in the order of hundreds of µm [11], micro-energy dispersive X-ray fluorescence spectrometry (µ-EDXRF) was used to investigate only the surface composition of the coins. Moreover, since no treatment of the sample was required, it was used to study the corrosion phenomena that may have occurred on their surface. All coins were analysed qualitatively by µ-EDXRF spectroscopy that allowed the identification of the main elements present on the surface of the denarii and antoniniani under study.

Although the spectra were not entirely overlapping in terms of peak intensity, all were confirmed on the presence of Ag, Cu, Sn, Ca, Fe, Pb, Au, Zn, Bi, and Ni (Figure 1). The higher concentration of copper and silver suggested the binary alloy nature of the coins, typically used by the Romans to mint coins during the 3rd century A.D. [13], while the presence of Fe and Ca is most likely due to the interaction between the coins and the burial soil components [3]. Subsequently, ICP-AES and ICP-MS micro-destructive analyses were carried out to quantify mainly these elements.

By calculating the area subtended by the peaks relative to the silver Kα1 line and to the copper Kα1 line, it was possible to calculate the ratio Ag/Cu in order to delineate the trend of the noble metal at the surface over the 2nd and 3rd centuries (Figure 2). An average decrease in silver can be observed, in accordance with the progression of the third-century crisis and the introduction of the antoninianus as a healing operation. In the case of the antoniniani issued by Gordian III, they showed a high surface inhomogeneity, which resulted in spectra with little overlap in intensity and consequently a high standard deviation.

#### 2.1.2. SEM-EDX Results

The SEM-EDX investigation was carried out on four well-preserved coins—two denarii (no. D31, D126) and two antoniniani (no. A48, A148)—in order to study the different distribution of elements on the coins’ surface. These samples were chosen because, for some coins, previous XRF analysis highlighted differences in the obtained spectra in terms of peaks’ intensity. Moreover, among these coins, the four samples subjected to SEM-EDX analysis were the best preserved and did not require pre-treatment for analysis. The EDX mapping (Figure 3) revealed an uneven metal distribution of Ag and Cu, and the presence of oxygen.

The uniform arrangement of oxygen over the entire studied area suggests that copper oxides have formed on the surface. Cuprite and tenorite are the main alteration compounds in Ag-Cu alloy artefacts. Indeed, as reported by Doménech et al. [14], the primary patina is composed of cuprite (Cu_2_O), while the patina associated to soil corrosion is mainly formed by tenorite (CuO). In some EDX spectra, the S and Cl peaks are well identified. This spectroscopic data supports the hypothesis that there may be acanthite (Ag_2_S) and chloroargyrite (AgCl) as insoluble corrosion compounds in the patina.

### 2.2. Quantitative Analyses on the Bulk

#### 2.2.1. ICP-AES Results

The normalised percentage concentrations—obtained by relating the mg/kg of the individual element to the sum of the ppm of all the elements detected and multiplying the ratio by 100—of the two main elements (Ag and Cu) in the bulk are shown in Figure 4. We reported the average value of concentration calculated on two replicates from the dilution, and the corresponding standard deviations. The analysis confirmed that the coins are based on a silver-copper alloy. Since the standard deviations had no statistical significance, we performed the *t*-test between the two currencies (%Ag content of denarii (N = 8) and %Ag content of antoniniani (N = 4)) and the result is that the silver concentrations of the two group of currencies are significantly different by 90% (ρ = 0.08).

Only one of the 12 coins analysed micro-destructively had a higher Cu content than Ag percentage (antoninianus no. A155), which was minted immediately later the Caracalla reform. At the opposite, the coin with the highest Ag content, approximately 76%, was the denarius no. D59: it is the only denarius that differs significantly from the others minted in the same period in terms of Ag percentage in alloy.

It is also worth noting the different relation between %Ag and %Cu of the two currencies: in the case of the denarii, minted before 215 A.D., there is an average difference of more than 10% between the % of Ag and that of Cu, while the antoniniani show almost equal percentages of the two major elements.

#### 2.2.2. ICP-MS Results

With regard to trace elements, Figure 5 shows the percentage elemental composition of the 12 coins analysed excluding Ag and Cu. Therefore, in this paragraph, the % of each trace element refers to the sum of trace elements concentration set equal to 100 (any % is then referred to the subgroup of trace elements, excluding Cu and Ag). Among the trace elements, the one with the highest percentage is Pb (average content 53.1 ± 2.6 wt%), in agreement with the literature about this element in silver coins [15,16]. In fact, in ancient times, this noble metal was extracted from galena (PbS), as this mineral contained appreciable amounts of silver combined as sulphur. Incomplete cupellation may leave Pb impurities in the Ag extracted [15]. The lead content is an indication of the effectiveness of the purification procedure. Looking at the overall data (Table 3), 5 of the 12 coins have a lead content greater than 0.5% indicating low grade silver. Bismuth is an impurity in argentiferous galena, which was often removed by the purification process: this agrees with the low concentrations found in the coins [17].

The high iron content in antoninianus no. A48 (51.7 ± 0.3 wt%) is peculiar. Since this coin was well preserved and the concentration is relative to the bulk of the sample, a correlation with the burial site can be excluded. Rather, this high iron content can be attributed to an incomplete copper refining process (probably due to some processing error that was then transferred onto a certain batch of coins); in ancient times, the most common copper ore used in metallurgy was, in fact, chalcopyrite (CuFeS_2_, which is the most abundant copper ore in the environment) [18].

The average Sn content of 21.3 ± 5.8% is a clear indication that copper in the alloy was introduced in the form of brass or bronze (with their minor elements) and not directly as copper (native or extracted but anyway with a high title). In fact, it is likely that during the 3rd century old scrap metal or worn coins from previous reigns were used for the debasement of denarii and antoniniani [2]. This is clearly detectable in the case of the antoninianus no. A155, where the highest Cu concentration (Figure 4) corresponds to the highest content of Sn (64.4 ± 0.2%, Figure 5).

### 2.3. Correlation Matrix and Chemometric Analysis of Data

We used the average physical data and the percentage of elements (Table 3) to calculate the correlation matrix in order to identify positive or negative correlations. We noted a negative correlation between diameter and thickness (−0.60) and a positive correlation between mass and diameter (0.76). Moreover, a negative correlation was observed between Ag and Cu (−1.0), underlining that these elements are the two components of the binary alloy that constituted the coins. Sn and Cu are positively correlated (0.56), as well as Ni and Cu (0.68), supporting the hypothesis of re-melted brass or bronze for coining, instead of pure copper [4]. On the other hand, Ag shows a positive correlation with Bi (0.82), as well as Pb and Bi (0.50), as foreseeable with the use of argentiferous galena as a mineral for silver extraction [17].

Principal component analysis (PCA) was performed on the same dataset used for the bivariate analysis, in order to identify terms of difference between the two currencies and their different features. Examination of plots obtained by PCA allows us to understand trends and patterns within samples, correlation among variables, and relationships between samples and variables [19].

Regarding correlation among variables, by studying the loadings plot it can be concluded that there is a positive correlation among Sn, Ni, Cu, and weight. A second group of positively interrelated variables consist of Ag, Pb, and Bi. A third group, characterized by positive interrelated variables, is formed by Zn, Ca, and thickness. Finally, Ag and Cu are located in an orthogonal direction in the PC1 versus PC2 plane.

From the investigation of the biplot (see Figure 6), we can notice that PC1 is mainly associated with elemental composition: on the right-hand side of the plot (at the highest PC1 score values) is the coin with the highest copper amount (coin no. A155), while at the lowest values (on the left) is the coin no. D59, the denarius that contained about 76% of Ag. PC2 mainly explains the physical characteristics: at the top of the plot (at the highest PC2 score values) are coins with higher diameter, such as antoniniani. Whereas, specimens with larger thickness, such as coins no. D23 and no. D130, are located at the bottom of the biplot.

Moreover, thanks to the biplot, the difference between the two currencies can be clearly identified. Two separate groups are defined: one consisting of coins no. A48, A159, and A160 which have a larger diameter, a higher relative amount of Cu, and a fair percentage of Fe (>0.5% for coin no. A48). The other group is located in the left-hand side of biplot and consists of seven denarii (coins no. D23, D27, D117, D125, D128, D130, and D151), which present a greater amount of lead and a higher average percentage of silver.

## 3. Materials and Methods

### 3.1. Chemicals and Apparatuses

All chemicals used were of analytical grade. Nitric acid (67–69% *v*/*v*) was from VWR (Milan, Italy). A Millipore purification pack system (MilliQ water, Millipore, Burlington, MA, USA) was used to produce water reagent grade. For the calibration curves for ICP-AES and ICP-MS analyses, the dilution of multi-standard solutions (10 mg/L) for ICP analyses was necessary (Periodic Table MIX 1 and Periodic Table MIX 2, Sigma-Aldrich, Saint-Louis, MO, USA). The elements examined were Ag, Cu, Pb, Bi, Ca, Fe, Ni, Zn, and Sn.

A Teccpo 170 W mini drill (8000–35,000 rpm) (Teccpo, Shenzen, China) was used in the sampling procedure with HSS drill bits in W (diameter 0.5–0.9 mm). The powder resulting from drilling was weighed by means of an analytical weigh scale, Chyo JL-180 (readability: 0.1 mg; standard deviation ≤0.02 mg; stabilization time: 5 s). Thanks to Vernier callipers, it was possible to measure diameter and thickness in three places on each coin.

### 3.2. µ-EDXRF

An ARTAX 200 micro-XRF spectrometer (supplied by Bruker Nano GmbH, Berlin, Germany) was used to perform the elemental analysis. The instrument has an air-cooled Mo X-ray fine focus tube (max 50 kV, 1 mA, 40 W) controlled by a compact high voltage generator unit and equipped with a 650 mm collimator. It has also a Peltier cooled XFlash^®^ silicon drift detector (10 mm^2^ of active area and energy resolution <150 eV for Mn–Ka at 100 kcps) and a CCD camera (500 × 582 pixels) for sample positioning. The focal spot is 1.2 × 0.1 mm^2^ with a 0.2 mm lateral resolution and a 100 μm beryllium window. For the hardware control and analytical data evaluation, the ARTAX control semi-quantitative XRF software was used (version 5.3.14.0, license of Bruker AXS Microanalysis GmbH, Berlin, Germany). Data are plotted as counts versus energy (keV). For our analyses, the instrumental parameters were set as follows: X-ray tube = 30 W, Mo target U = 50 kV, I = 700 μA, acquisition time = 45 s (live time), collimator = 650 mm (air environment). The examined elements were: Ag (line: Lα1 2.98 keV; Kα1 22.16 keV), Cu (line: Kα1 8.05 keV), Pb (line: Lα1 10.55 keV; Lβ1 12.61 keV), Zn (line: Kα1 8.64 keV), Fe (line: Kα1 6.40 keV), Ni (line: Kα1 7.48 keV), Sn (line: Kα1 25.27 keV), and Ca (line: Kα1 3.69 keV).

Two spectra were collected on the recto and two on the verso of each coin, making a total of four spectra for each specimen in order to obtain representative data of the entire coin.

### 3.3. SEM-EDX

Scanning electron microscopy with energy dispersive X-ray spectroscopy (SEM-EDX) measurements were performed on 4 coins in order to study the superficial inhomogeneity of the samples. A Carl Zeiss Sigma VP (Sigma-Aldrich, Saint-Louis, MO, USA) field emission scanning electron microscope (FE-SEM) equipped with a Bruker Quantax 200 microanalysis detector was used to carry out the analyses. SEM micrographs, EDX spectra, and maps were recorded under the same conditions (20 keV, 60 µm condenser aperture) for all the samples; both secondary electrons and backscattered electrons were used as imaging signals.

### 3.4. Sample Preparation

Sampling is required for ICP-AES and ICP-MS analyses. The material was taken using a mechanical drill equipped with a tungsten-carbide drill bit (0.8 mm diameter at 10,000 rpm). It was decided to drill the edge of coin and proceed along the radius to a depth of about 5 mm, so as not to affect the coins’ faces. The powder drilled (about 15 mg) was collected in weighing boats in polystyrene and weighed, dissolved in a concentrated (67–69% *v*/*v*) nitric acid aqueous solution, and heated overnight to facilitate the dissolution. To remove suspended particles, a sonication step was necessary. Lastly, solutions were diluted in a 100 mL calibrated flask (class A) with MilliQ water.

### 3.5. ICP-AES

The major elements (Ag and Cu) were determined by analysing sample solutions with an ICP-AES spectrometer (PerkinElmer^®^ Optima™ 8000, Waltham, MA, USA). Calibration (linear in the concentration range of 0.1–10 mg/L) was performed after the dilution of a multi-standard solution 10 mg/L for ICP analysis (Periodic table mix 1 for ICP TraceCERT^®^, Sigma-Aldrich). The limits of detection (LOD) in the solution sample at the operative wavelength for each element were: 0.02 mg/L for Cu at 327.393 nm and 0.02 mg/L for Ag at 328.068 nm. The coefficients of variation of repeatability (RSD %) were found to be <5%.

### 3.6. ICP-MS

Trace metals concentrations—Pb, Bi, Ca, Fe, Ni, Zn, and Sn—were determined by inductively coupled plasma-mass spectrometry (ICP-MS) using a NexION 350x Spectrometer (PerkinElmer, Waltham, MA, USA) equipped with an ESI SC autosampler. In order to control and minimize cell-formed polyatomic ion interference, the analysis was performed in KED mode (kinetic energy discrimination) using ultra-high purity helium (flow rate of 4.8 mL/min). The calibration of instrument (linear in the concentration range of 0.5–100 µg/L) was carried out following dilution of a multi-standard solution 10 mg/L for ICP analysis (Periodic table mix 1 for ICP TraceCERT^®^ and Periodic table mix 2 for ICP TraceCERT^®^, Sigma-Aldrich). The measurements of samples were performed using the calibration curve method obtained by analysing standard solutions. The limits of detection (LOD) for each element were: Pb 0.005 µg/L; Bi 0.001 µg/L; Ca 1.5 µg/L; Fe 0.7 µg/L; Ni 0.1 µg/L; Zn 0.2 µg/L; and Sn 0.01 µg/L. The coefficients of variation of repeatability (RSD %) were <3%.

### 3.7. Data Processing

Bivariate analysis was performed by computation of Pearson’s correlation coefficients between couples of variables, selected from the physical and chemical data. Computations were performed using MS-Excel (Microsoft Corporation, Redmond, WA, USA, Version 2019). The same data were analysed by principal component analysis (PCA), too. PCA is an unsupervised exploratory chemometric tool for the identification and visual representation of relationships: (i) among the samples (PC scores and score plots), (ii) within variables (PC loadings and loading plots), and (iii) between samples and variables (biplots) [20]. Due to the different nature and different measurements units of data, pre-processing of the data matrices (column-autoscaling) was necessary before the multivariate analysis [21]. Multivariate data processing was performed using the CAT (Chemometric Agile Tool, update of September 13, 2022) package, based on the R platform (R version 3.1.2, Copyright (C) 2014 The R Foundation for Statistical Computing, Vienna, Austria) and freely distributed by Gruppo Italiano di Chemiometria (Italy) [22].

## 4. Conclusions

For the 12 coins subjected to micro-destructive analysis, a combination of µ-EDXRF, SEM-EDX, ICP-AES, and ICP-MS techniques allowed the determination of the alloy percentage (dominant elements) and the trace elements concentrations of ancient denarii and antoniniani. The multi-analytical approach enables to unveil the chemical composition of ancient coins and to speculate on raw materials’ nature and minting processes, as well as to identify surface corrosion compounds (which helps to address the hypotheses on the environmental history of the finds).

The presence of Ag, Cu, Pb, Sn, Ni, Zn, Bi, Fe, and Ca and the inhomogeneity of the coins’ surfaces can be deduced from the responses of the non-destructive analysis conducted on all 160 finds. Also considering the surface roughness that may affect the response of the XRF analysis, slight irregularities in the trends and imperfect superimposition of the spectra are indications of an inhomogeneous surface chemical composition. The different distribution of the two main alloy’s elements was also highlighted by SEM-EDX analysis, carried out on four specimens (two denarii and two antoniniani).

Due to the great value of these historical specimens, micro-destructive analyses involved only 12 coins. In this study, a few milligrams of sample were picked up at a depth of a few millimetres. The sampling method adopted, despite some limitations mainly due to the scarcity of the available sample, provided representative samples of inhomogeneous ancient coins and allowed us to measure the elemental composition (major and minor elements) of the coins under study. We discovered that the alloy composition of the eight studied denarii is almost constant, except for denarius no. D59, which has a higher concentration of silver, and denarius no. D130, which has almost equal concentrations of Cu and Ag. The four antoniniani analysed also have a similar alloy percentage, except for antoninianus no. A155, which presents a greater concentration of Cu. However, we have clearly noticed the difference between the two currencies, with a net decrease in the % of Ag as the year of minting increases, underlining the progression of the third century economic crisis.

Among the trace elements, the high concentrations of Pb and Sn are in agreement with the use of argentiferous galena for silver extraction and old brass or bronze objects for the production of the silver-copper coins, respectively. This was confirmed also by bivariate and multivariate correlations. The latter statistical investigation identified the two groups of coins corresponding to the two currencies, providing a first basis for a numismatic database, and allowing the chemical mapping of unknown samples.

## Figures and Tables

**Figure 1 molecules-27-06903-f001:**
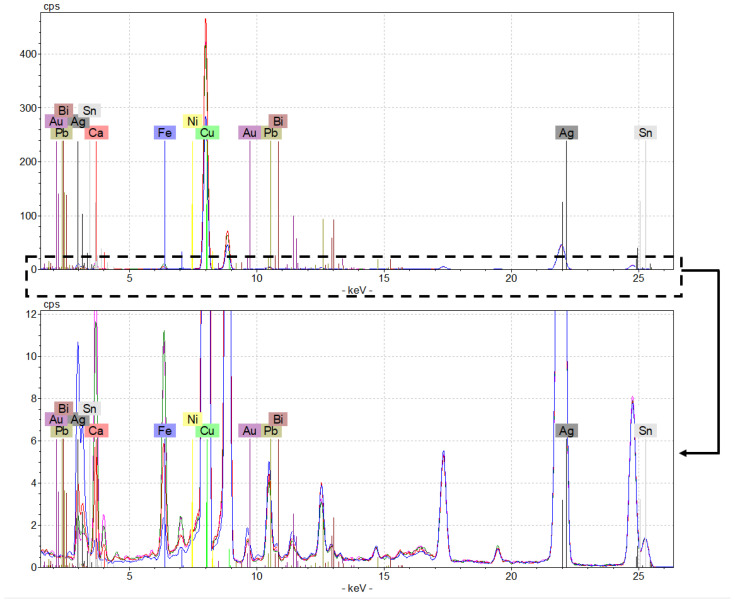
Four overlapped µ-EDXRF spectra of the coin no. D27. Spectrum of recto_1 in pink, recto_2 in blue, verso_1 in red, and verso_2 in green. The lower spectrum is relative to the area within the dashed frame, so that even the lower intensity peaks can be appreciated.

**Figure 2 molecules-27-06903-f002:**
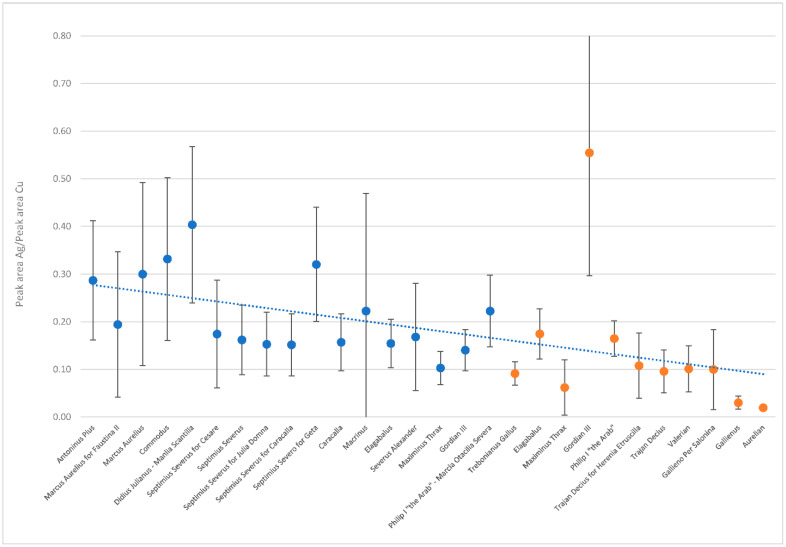
Average Ag/Cu ratio of the 160 coins analysed with respective error bars, denarii in blue and antoniniani in orange. The x-axis shows the emperors who issued the coins under study. The ratio reported in y-axis regards µ-EDXRF signals.

**Figure 3 molecules-27-06903-f003:**
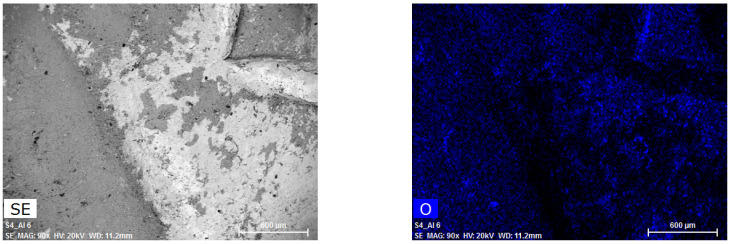
SE image and X-ray maps on the surface of antoninianus no. A48 obtained by SEM-EDX.

**Figure 4 molecules-27-06903-f004:**
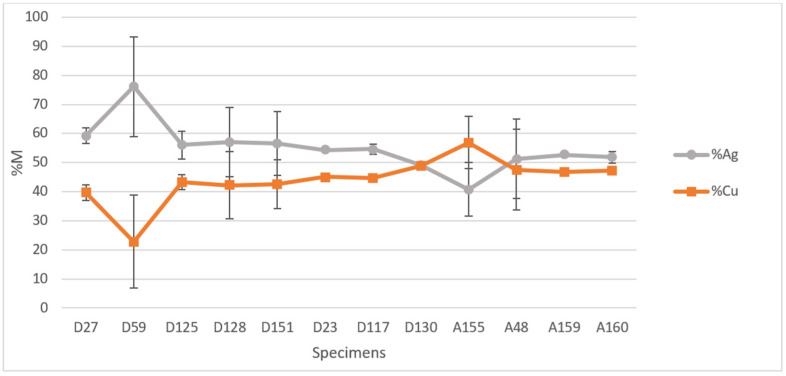
Relative concentration of Ag and Cu in the 12 coins (8 denarii and 4 antoniniani) analysed by ICP-AES. Data are given as mean ± RSD%.

**Figure 5 molecules-27-06903-f005:**
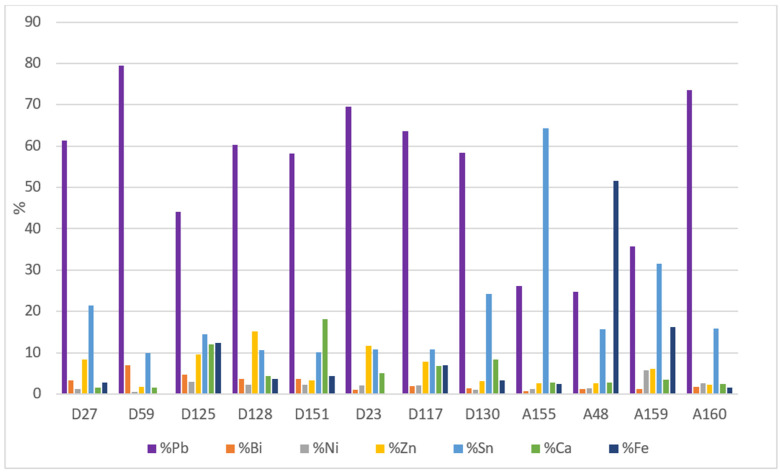
Average content of trace metals in the bulk of the 12 coins analysed by ICP-MS. The data are normalised excluding Ag and Cu content.

**Figure 6 molecules-27-06903-f006:**
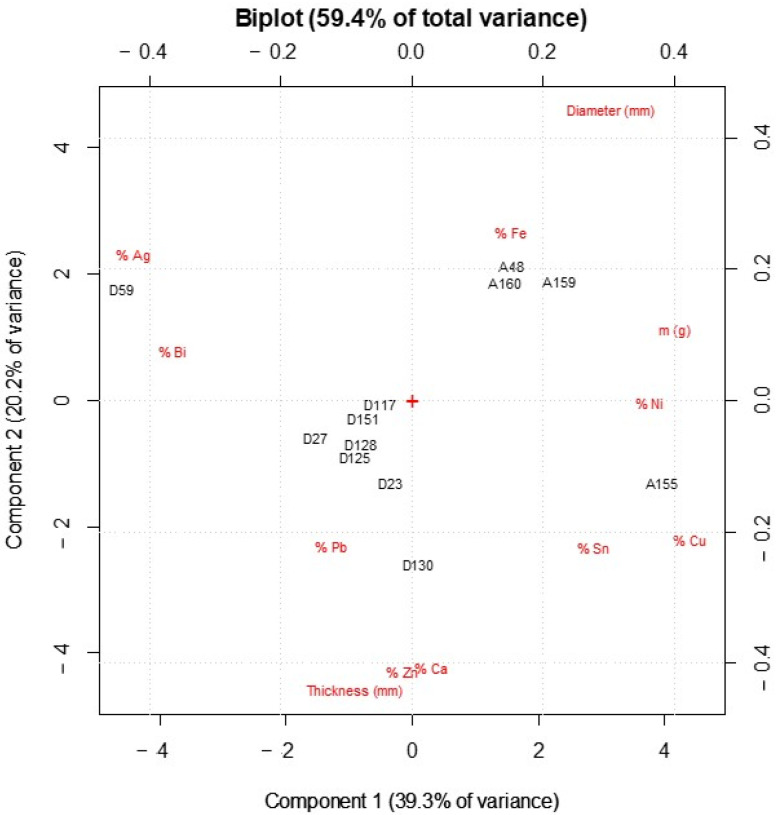
Biplot (score and loading scatter diagram) from PCA: scores are in black, loadings in red.

**Table 1 molecules-27-06903-t001:** List of coins under study. Each coin is associated with a code where A = antoninianus, D = denarius, followed by a progressive number from 1 to 160. Coins marked with a star (*****) are sampled for ICP-AES and ICP-MS analyses, while coins marked with a cross (^†^) are those analysed with SEM-EDX.

Emperor	Period of Issue (A.D.)	Number of Samples	Sample Code
Antoninus Pius	138–161	1	D144
Antoninus Pius for Diva Faustina I	141–161	1	D35
Marcus Aurelius for Faustina II	145–161	3	D43; D65; D137
Marcus Aurelius	161–180	3	D38; D112; D116
Commodus	177–192	7	D2; D15; D54; D67; D78; D93; D108
Didius Julianus– Manlia Scantilla	193	1	D121
Septimius Severus	193–211	54	D1; D3; D4; D9; D11; D13; D14; D29; D30; D31 ^†^; D33; D37; D40; D42; D45; D46; D50; D51; D52; D53; D55; D57; D58; D59 *; D60; D61; D63; D64; D66; D71; D72; D73; D74; D76; D80; D82; D84; D86; D87; D97; D98; D99; D100; D110; D113; D118; D119; D122; D128 *; D129; D138; D140; D143; D151 *
Septimius Severus for Julia Domna	193–211	24	D5; D8; D17; D19; D21; D27 *; D28; D34; D41; D47; D49; D56; D70; D77; D95; D106; D114; D115; D123; D125 *; D131; D132; D141; D142
Septimius Severus for Caracalla	198–209	13	D6; D23 *; D25; D39; D44; D62; D75; D85; D89; D117 *; D130 *; D134; D149
Septimius Severus for Caesar	198	1	D7
Septimius Severus for Geta	209–212	2	D10; D96
Caracalla	198–217	8	D12; D18; D68; D79; D94; D102; D107; D145
Trebonianus Gallus	215–253	1	A150
Macrinus	217–218	1	D32
Elagabalus	218–222	13	D36; D69; D83; D92; D104; D105; D126 ^†^; D133; D136; D139; A147; A152; A155 *
Severus Alexander	221–235	3	D26; D90; D120
Maximinus Thrax	230–235	3	D103; D157; A20
Gordian III	238–244	6	D91; D153; A48 *^,†^; A158; A159 *; A160 *
Philip the Arab	244–249	4	A81; A109; A111; A154
Philip the Arab– Marcia Otacilia Severa	244–249	1	D127
Trajan Decius	249–251	2	A101; A135
Trajan Decius for Herenia Etruscilla	249–251	2	A22; A24
Valerian	253–260	3	A88; A146; A156
Gallienus	260–268	1	A148 ^†^
Gallienus for Salonina	260–268	1	A16
Aurelian	270–275	1	A124

**Table 2 molecules-27-06903-t002:** Recto and verso of the 12 coins micro-sampled.

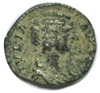	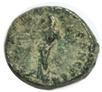	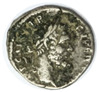	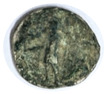	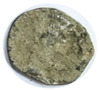	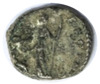
D27, Septimius Severus for Julia Domna	D59, Septimius Severus	D125, Septimius Severus for Julia Domna
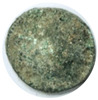	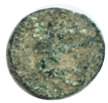	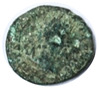	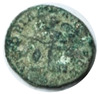	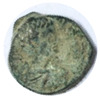	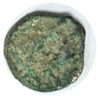
D128, Septimius Severus	D151, Septimius Severus	D23, Septimius Severus for Caracalla
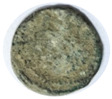	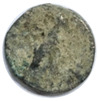	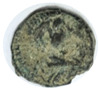	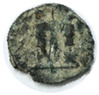	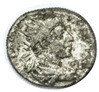	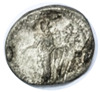
D117, Septimius Severus for Caracalla	D130, Septimius Severus for Caracalla	A155, Elagabalus
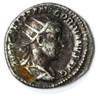	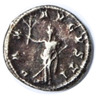	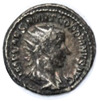	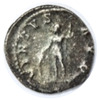	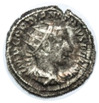	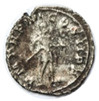
A48, Gordian III	A159, Gordian III	A160, Gordian III

**Table 3 molecules-27-06903-t003:** Average concentrations in percentage (wt%) and mean physical data for the 12 coins subjected to micro-destructive analysis.

Sample	Year (A.D.)	Mass (g)	Diameter (mm)	Thickness (mm)	% Ag	% Cu	% Pb	% Sn	% Bi	% Zn	% Ni	% Ca	% Fe
D27	193–211	3.4587	16.5	2.4	59.3	39.7	0.630	0.219	0.033	0.085	0.013	0.017	0.028
D59	193–211	3.3158	17.1	2.2	76.1	22.8	0.871	0.109	0.076	0.018	0.005	0.017	<LOD *
D125	193–211	3.4548	15.3	2.5	56.1	43.3	0.278	0.092	0.030	0.060	0.018	0.076	0.077
D128	193–211	3.5548	17.3	2.3	57.0	42.2	0.451	0.080	0.028	0.113	0.017	0.033	0.027
D151	193–211	3.5763	16.5	2.3	56.6	42.7	0.395	0.069	0.025	0.022	0.015	0.123	0.029
D23	198–209	4.0493	16.3	2.8	54.3	45.1	0.402	0.062	0.006	0.067	0.012	0.029	<LOD
D117	198–209	3.5017	15.9	2.2	54.6	44.7	0.392	0.066	0.011	0.049	0.013	0.042	0.043
D130	198–209	3.4990	15.6	2.3	49.1	48.9	1.193	0.495	0.028	0.063	0.023	0.171	0.069
A155	218–222	5.6444	19.9	2.4	40.8	56.9	0.598	1.470	0.014	0.059	0.026	0.063	0.055
A48	238–244	4.4082	20.8	2.2	51.3	47.6	0.269	0.171	0.014	0.029	0.014	0.031	0.564
A159	238–244	4.4958	21.5	2.0	52.7	46.7	0.219	0.194	0.008	0.037	0.035	0.022	0.100
A160	238–244	4.5434	21.6	1.9	51.8	47.3	0.669	0.145	0.016	0.020	0.023	0.023	0.015

* LOD = Limit of Detection.

## Data Availability

The data presented in this study are available on request from the corresponding author.

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
