# Peer review of "A Multi-Analytical Approach on Silver-Copper Coins of the Roman Empire to Elucidate the Economy of the 3rd Century A.D."

_molecules, 2022, doi:10.3390/molecules27206903_

Round 1

Reviewer 1 Report

Manuscript Number: Molecules-1950297

The manuscript “A multi-analytical approach on silver-copper coins of the Roman empire to reconstruct the second half of the 3rd century A.D. economy” presents study of surface of 160 coins using mEDXRF, 12 samples were analyzed by ICP/AES and ICP/MS, and 4 samples were investigated by SEM/EDX.

The topic is interesting; it is suitable for this journal and obtained results are valuable.

I would recommend this manuscript to be published in Molecules after following minor revisions:

-          Better description of samples is needed: at least a table where all subgroups of 160 samples will be defined, including dating and number of samples per each subgroup.

-          Sample marks should be explained, e.g. noD31, D126….

-          Explain criteria how 12 samples for drilling were chosen.

-          Explain why only 4 samples were  investigated by SEM/EDX- criteria of choice.

-          Authors define three different aims of the work: lines 19-21, lines 41-43, and lines 109-112. None of stated aims were accomplished. It should be rewritten carefully.

-          Line 40, define “which period”.

-          Lines 63-64, “similar currencies (antoniniani)” - it is just plural of antoninianus

-          Lines 202-205, no subject in two sentences; add “coins”

-          Line 203, “surface composition” add of what

-          Line 206, “all agreed” change to “all confirmed”

-          Lines 208-2012, show results to support these conclusions, e.g. XRF spectra of representative samples

-          Line 271, cite appropriate literature.

-          Line 344, support claim of inhomogeneity of coins with the results

-          Why chemometric analysis was not performed on XRF results obtained on the surface of 160 coins and compared with presented results on 12 coins (not large assemblage for statistical analysis)?

Typos:

1.      Line 337, mXRF

Reviewer 2 Report

The paper "A multi-analytical approach on silver-copper coins of the Roman empire to reconstruct the second half of the 3rd century A.D. economy" is an exciting study of the Roman coins in efforts to obtain more data on the historical period of Barracks emperors (and the previous years). Authors applied both non-destructive and desctructive techniques to gain information about chemical composition of ingots used for the coins minting. The paper comes handy for historicists and other specialists interested in the civilizations of the past.

I have only a couple of minor comments:

1. lines 250-251. There is no wonder that R2 = 0.996; the dependence of x on y when y = 1-x is doomed to have R2 = 1. In my opinion, the conclusion "this suggests that copper was intentionally added to the silver" does not follow from the obvious fact of linear interdependence between components' content in the binary system.

2. Actually, from Fig. 3, there is no clear variation in Ag content if the error bars are taken into account. At least, it should be proven with t-test.

3. Fig. 1. It would be nice to add an information about the sample size; without it, the error bars are not that informative.

4. If I understood correctly, the conclusion of the paper is "The situation in the Empire went from bad to worse because they had to add more and more copper to silver and re-melt brass" and it refers to the economy. Is that all? If so, the promise to "reconstruct the second half of the 3rd century A.D. economy" given in the title seems, you know, an exaggregation un po'.
